# Robust Object Detection with Pseudo Labels from VLMs using Per-Object Co-teaching

## Abstract

Foundation models, especially vision-language models (VLMs), offer compelling zero-shot object detection for applications like autonomous driving, a domain where manual labelling is prohibitively expensive. However, their detection latency and tendency to hallucinate predictions render them unsuitable for direct deployment. This work introduces a novel pipeline that addresses this challenge by leveraging VLMs to automatically generate pseudo-labels for training efficient, real-time object detectors. We extend the per-object coteaching-based training strategy to the YOLO family of models using anchor level selection and a selection loss that excludes objectness score. Per-object coteaching filters noisy bounding boxes from training instead of filtering the entire image. Specifically, two YOLO models learn collaboratively, filtering out unreliable boxes from each mini-batch based on their peers' per-object loss values. Overall, our pipeline provides an efficient, robust, and scalable approach to train high-performance object detectors for autonomous driving, significantly reducing reliance on costly human annotation. Experimental results on the KITTI dataset demonstrate that our method outperforms a baseline YOLOv5m model, achieving a significant mAP@0.5 boost (31.12% to 46.61%) while maintaining real-time detection latency. Furthermore, we show that supplementing our pseudo-labelled data with a small fraction of ground truth labels (10%) leads to further performance gains, reaching 57.97% mAP@0.5 on the KITTI dataset. We observe similar performance improvements for the ACDC and BDD100k datasets.

## 1 Introduction

Real-time object detection is paramount for safe navigation in autonomous driving systems, demanding rapid and accurate environmental perception. Traditional object detection methods, while effective, rely on extensive and precise human-annotated data, which is both labour and capital-intensive. Vision-Language Models (VLMs) have emerged as a promising alternative, demonstrating remarkable zero-shot detection capabilities for a broad range of objects described through natural language prompts. This enables a potentially scalable paradigm where detector performance is no longer limited by the availability of human-labelled data Zhao et al. (2022).

However, deploying large-scale VLMs directly in real-time autonomous driving scenarios faces significant hurdles. First, the pseudo-labels generated by VLMs are often noisy and imprecise, particularly in challenging edge cases like occlusions or adverse weather conditions, making them unreliable for safety-critical applications Li et al. (2023). Second, VLMs are computationally expensive, rendering them impractical for real-time inference on resource-constrained automotive platforms Shinde et al. (2025). Simply training fixed-task object detectors on these pseudo-labels can lead to significant performance degradation due to inaccurate bounding boxes and misclassified objects Singh et al. (2024). Thus, a key challenge lies in developing methods to mitigate the noise inherent in VLM-generated labels and extract a reliable training signal Li et al. (2020b).

To address these challenges, we propose a novel pipeline that combines the benefits of VLM-based pseudo-annotation with a robust per-object-based co-teaching training strategy. Our approach leverages the zero-shot knowledge of VLMs to generate pseudo-labels and then trains two randomly initialised YOLOv5 models

simultaneously. Each model selectively filters out potentially noisy samples from each mini-batch based on the other model's loss values. This allows us to leverage the scalability of VLMs while effectively mitigating the impact of inaccurate pseudo-labels. Notably, our approach is designed to outperform vanilla model distillation, which is negatively influenced by noisy teacher labels Li et al. (2020b), and benefits significantly from the inclusion of even a small percentage of ground truth data Tang et al. (2021).

We make the following key contributions in this paper.

- We propose a per-object coteaching-based pipeline to generate pseudo labels from VLMs and train robust, efficient student models on the labels.

- We extend the per-object coteaching approach from Chadwick & Newman (2019) to the YOLO family of models with anchor-based selection and a selection loss for sorting objects.

- Our approach outperforms a baseline YOLOv5 model trained on raw pseudo-labels. Detailed experimental analysis to demonstrate a significant rise in the detection performance on KITTI, ACDC and BDD100K datasets. More specifically, we observe that the mAP@0.5 score improves by 15.49% on the KITTI dataset, 7.19% on the ACDC dataset and 11.07% on the BDD100K dataset.

- We perform a detailed ablation study to further analyse our pipeline in various settings.

Additionally, the proposed approach is computationally efficient compared to direct VLM inference, as YOLO is a single-stage efficient detection model and suitable for real-time object detection. This pipeline leverages unlabeled data without reliance on human annotation, which makes it more scalable.

## 2 Related Work

### 2.1 Object Detection

**Open Vocabulary Detectors**  Zero-shot object detection addresses the challenge of detecting objects from categories not seen during training. Open-vocabulary object detection  Gu et al. (2022); Minderer et al. (2022) expands this concept by allowing detection models to identify objects based on natural language descriptions without explicitly being trained on these classes.

Foundation models like OWL Minderer et al. (2022) and OWLv2 Minderer et al. (2024) leverage pre-trained vision-language models to enable zero-shot detection capabilities. These models align visual and textual embeddings in a shared semantic space, allowing the detection of objects described by arbitrary text prompts without category-specific training data.

OWLv2 Minderer et al. (2024) builds upon the original OWL architecture with improved training strategies and a more efficient design. It uses a vision transformer (ViT) backbone combined with a text encoder to process image regions and textual descriptions, computing similarity scores between them. This makes OWLv2 particularly valuable as an auto-labeller for domains with limited labelled data or novel object categories—a common scenario in autonomous driving environments.

Although foundation models like OWLv2 offer powerful zero-shot capabilities, they typically have substantial computational requirements that make them impractical for direct deployment on autonomous vehicles with limited hardware resources and real-time processing constraints Minderer et al. (2024); Zhu et al. (2022).

**Single-Stage Detectors**  Single-Stage Detection methods, particularly the YOLO family of models, are popular for real-time applications. YOLO ("You Only Look Once") Redmon et al. (2016) pioneered a one-pass detection architecture that predicts bounding boxes and classes in a single network forward pass. This was followed by multiple updates to (v2 Redmon & Farhadi (2017), v3 Redmon & Farhadi (2018), v4 Bochkovskiy et al. (2020), v5 Jocher (2020), v7 Wang et al. (2022), v8 Jocher et al. (2023), etc.) which focused on improving performance while maintaining or improving latency. Recently Cheng et al. (2024) combined YOLO's efficiency with open-vocabulary capabilities using vision–language pre-training and a region-text contrastive loss to detect a wide range of object classes in a zero-shot manner.

## 2.2 Learning with Noisy Labels

Training neural networks with noisy labels is a challenging task because the networks can eventually fit the noise. Methods like MentorNet Jiang et al. (2018) proposed learning a curriculum model to down-weight or discard examples suspected to have wrong labels. Coteaching Han et al. (2018) is a training paradigm proposed to mitigate label noise where two identical models with random initialisation are trained in parallel, selecting a subset of small-loss (likely clean) examples from each mini-batch for the other network to learn from. It was further extended with some improvements in the classification setting Yu et al. (2019). Numerous extensions and alternatives have since been explored. Overall, the literature shows that tolerating or filtering noise during training (through co-teaching, mentor models, robust loss functions, etc.) is vital for maintaining performance when learning with noisy labels. We build on these insights to handle errors in pseudo-box annotations.

## 2.3 Pseudo-Labelling Strategies for Object Detection

Using pseudo-labels (model-predicted labels on unlabeled data) is a key technique in object detection. In self-training, a teacher model's detections on unlabeled images are treated as ground truth to train a student model. Radosavovic et al. (2018) is an early approach towards omni-supervised learning for object detection. It generates pseudo-bounding boxes by ensembling a model's predictions under multiple image transformations and then retraining the detector on this augmented pseudo-labelled set. A critical factor in this direction is filtering out poor predictions to avoid overfitting to bad data. Recent semi-supervised frameworks address this by using confidence thresholds or teacher-student mutual learning. Liu et al. (2021) is a notable method that mitigates bias toward easy classes in pseudo-labels. The recent rise in foundation models Minderer et al. (2022) enables generating strong pseudo labels from an open set of labels. Zhao et al. (2022) computes the Grad-CAM activation map over cross-attention in VLMs and picks the proposal box that best overlaps the activation to generate pseudo labels. Gao et al. (2022) generates class-agnostic region proposals, improves its localisation, and classifies each box with CLIP and post-processes these labels with NMS and confidence thresholding to generate pseudo labels. However, these pseudo-labels generated by open vocabulary models contain noisy labels, hallucinated boxes and inaccurate box coordinates.

## 2.4 Robust Object Detection

Training robust object detectors on noisy data goes beyond noisy labels. These methods have to deal with label noise, missing annotations, inaccurate bounding box coordinates, or out-of-distribution inputs. Chadwick & Newman (2019) systematically analysed how different noise types (classification errors, localisation errors, etc.) affect object detection. They proposed a per-object co-teaching strategy to mitigate label noise while training an R-CNN. Our approach differs in filtering strategy, which is based on the YOLO loss function. Li et al. (2020a) proposed an alternating optimisation scheme that iterates between correcting noisy labels and updating the detector. This handles noise in both class labels and box coordinates. Wan et al. (2019) introduced a meta learning solution using a small set of trusted, clean samples. Liu et al. (2022) proposed an R-CNN framework focusing on training with inaccurate bounding box coordinates.

# 3 Preliminaries

This section provides an overview of the key concepts and prior work foundational to our approach: zero-shot object detection, YOLO based single stage detection methods and coteaching-based methods to train robust models on noisy data.

## 3.1 Object Detection with YOLO

Object detection is a fundamental computer vision task that involves localising and classifying objects within an image. For autonomous driving applications, object detection models must balance accuracy with real-time performance to ensure safe navigation. The You Only Look Once (YOLO) Redmon et al. (2016);

Redmon & Farhadi (2018) family of models have emerged as a leading approach for real-time object detection by framing detection as a regression problem.

YOLOv5 Jocher et al. (2021) represents a significant advancement in the YOLO architecture, offering various model sizes (from nano to extra-large) that provide different efficiency-accuracy trade-offs. The architecture divides an input image into a grid and predicts bounding boxes and class probabilities directly from full images in a single evaluation. The loss function in YOLOv5 is composed of three primary components:

$$\mathcal{L} = \lambda_{coord}\ell^{\text{box}} + \lambda_{obj}\ell^{\text{obj}} + \lambda_{cls}\ell^{\text{cls}} \tag{1}$$

Where $\ell^{\text{box}}$ represents the bounding box regression loss (typically a combination of CIoU loss Zheng et al. (2020)), $\ell^{\text{obj}}$ is the objectness confidence loss, and $\ell^{\text{cls}}$ is the classification loss. The $\lambda$ terms are weighting factors that balance the contributions of each component.

While YOLO models provide efficient inference, they typically require extensive labelled training data and struggle with novel or rare object categories—a significant limitation for autonomous driving in complex and unpredictable real-world environments.

### 3.2 Learning with Noisy Labels using Coteaching

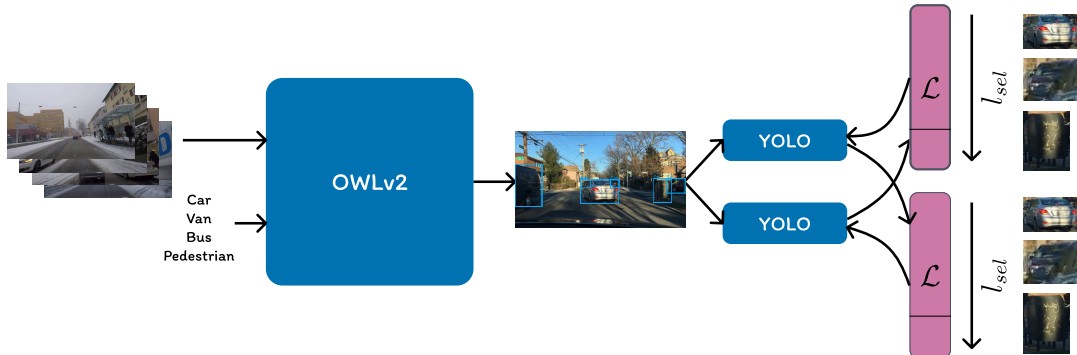

Figure 1: Pipeline for training robust, open vocabulary, real-time object detectors

**Co-teaching.** Co-teaching Han et al. (2018) is a robust training strategy designed to mitigate the impact of noisy labels. It employs two networks trained simultaneously, with each network learning from a different subset of the data. At every training iteration, each network selects samples from the mini-batch that produce the smallest losses (presumably clean labels) and uses these samples to update its peer network. The key intuition behind co-teaching is that low-loss samples are more likely to be correctly labelled, allowing models to mutually reduce the influence of noisy data. Formally, given two networks $f_1$ and $f_2$ with parameters $\theta_1$ and $\theta_2$, each network selects a proportion $R(T)$ of samples with the smallest losses. The updates are performed as follows:

$$\theta_1^{t+1} = \theta_1^t - \eta\nabla\mathcal{L}(f_1(X_{\text{small}}^{(2)}), Y_{\text{small}}^{(2)}) \tag{2}$$

$$\theta_2^{t+1} = \theta_2^t - \eta\nabla\mathcal{L}(f_2(X_{\text{small}}^{(1)}), Y_{\text{small}}^{(1)}) \tag{3}$$

where $X_{\text{small}}^{(i)}$ and $Y_{\text{small}}^{(i)}$ represent samples selected by network $i$ based on their lowest losses, and $\eta$ is the learning rate.

---

**Algorithm 1:** Per-Object Co-Teaching for Robust YOLO Training with Pseudo-Labels

---

**Input:** Training images $\mathcal{X}_{\text{tr}}$, prompts $\mathcal{P}$; Test set $(\mathcal{X}_{\text{te}}, \mathcal{Y}_{\text{te}})$;
Estimated noise rate $\hat{r}$; total epochs $T$; ramp-up epochs $T_k$

**1 Pre-processing:** Obtain pseudo-labels using an open-vocabulary detector (OVD): $\tilde{\mathcal{Y}}_{\text{tr}} \leftarrow \text{OVD}(\mathcal{X}_{\text{tr}}, \mathcal{P})$;

**2** From the noisy training set $\tilde{\mathcal{D}}_{\text{tr}} = \{(x_i, \tilde{y}_i)\}$;

**3 Initialise** two YOLO models $f_\theta$ and $g_\phi$ with random weights;

**4 for** $e = 1$ **to** $T$ **do**

**5**     $r_e \leftarrow \hat{r} \cdot \min(e/T_k, 1)$;

**6**     **foreach** *mini-batch* $\mathcal{B} = \{(x_b, \tilde{y}_b)\}_{b=1}^{B} \subset \tilde{\mathcal{D}}_{\text{tr}}$ **do**

**7**        **Forward**;

**8**        $P_f \leftarrow f_\theta(\mathcal{B})$;

**9**        $P_g \leftarrow g_\phi(\mathcal{B})$;

**10**        **Anchor-level selection loss (per positive anchor** $j$**)**;

**11**        $\ell_{f,j}^{\text{sel}} = \lambda_{\text{box}}\ell_{f,j}^{\text{box}} + \lambda_{\text{cls}}\ell_{f,j}^{\text{cls}}$;

**12**        $\ell_{g,j}^{\text{sel}} = \lambda_{\text{box}}\ell_{g,j}^{\text{box}} + \lambda_{\text{cls}}\ell_{g,j}^{\text{cls}}$;

**13**        $N_{\text{pos}} \leftarrow$ # positive anchors in $\mathcal{B}$;    $k \leftarrow \lceil (1 - r_e)N_{\text{pos}} \rceil$;

**14**        **Co-teaching filter**;

**15**        $\mathcal{K}_f \leftarrow$ indices of the $k$ smallest $\{\ell_{g,j}^{\text{sel}}\}$;    $\mathcal{K}_g \leftarrow$ indices of the $k$ smallest $\{\ell_{f,j}^{\text{sel}}\}$;

**16**        **Masked YOLO loss**;

**17**        $\mathcal{L}_f = \sum_{j \in \mathcal{K}_f} \left(\lambda_{\text{box}}\ell_{f,j}^{\text{box}} + \lambda_{\text{cls}}\ell_{f,j}^{\text{cls}} + \lambda_{\text{obj}}\ell_{f,j}^{\text{obj}}\right)$;

**18**        $\mathcal{L}_g = \sum_{j \in \mathcal{K}_g} \left(\lambda_{\text{box}}\ell_{g,j}^{\text{box}} + \lambda_{\text{cls}}\ell_{g,j}^{\text{cls}} + \lambda_{\text{obj}}\ell_{g,j}^{\text{obj}}\right)$;

**19**        **Back-propagation**;

**20**        $\theta \leftarrow \theta - \eta \nabla_\theta \mathcal{L}_f$;

**21**        $\phi \leftarrow \phi - \eta \nabla_\phi \mathcal{L}_g$;

**Output:** Trained parameters $(\theta, \phi)$ of two YOLO detectors $f_\theta$ and $g_\phi$

**22 Inference:** $\widehat{\mathcal{Y}}_f \leftarrow f_\theta(\mathcal{X}_{\text{te}})$;

**23**     $\widehat{\mathcal{Y}}_g \leftarrow g_\phi(\mathcal{X}_{\text{te}})$;

---

## 4 Methodology

In this section, we describe our methodology for building efficient and robust object detection models for autonomous driving. Our approach combines training on foundation model outputs with per-object coteaching to create lightweight and robust detectors that can operate in challenging real-world conditions and scale with an increasing stream of unlabelled data.

Our pipeline consists of three main components:

1. A foundation model (OWLv2) that serves as an auto-labeller for an open set of classes.

2. Two student YOLO models that train on the outputs from the teacher.

3. A per-object coteaching mechanism that enables the student models to discard the noisy objects and train on clean samples.

Figure 1 illustrates the overall architecture of our approach.

### 4.1 Generating Pseudo Labels Using Foundation Model (VLM)

We employ OWLv2 Minderer et al. (2024) as our foundation model teacher. OWLv2 is a vision-language model that excels at open-vocabulary object detection, allowing it to identify and localise a wide range of

objects beyond those seen during training. This capability is crucial for autonomous driving, where vehicles must recognise and respond to unusual or rare objects.

While OWLv2 provides high-quality detections, it has two limitations that we address:

- Computational overhead: OWLv2 is too large and slow for real-time inference on automotive hardware.

- Label noise: Foundation models can produce hallucinated or inaccurate detections, especially in edge cases.

We use OWLv2 offline to generate pseudo-labels on a large, diverse dataset of driving scenarios. These pseudo-labels serve as the foundation for training our student models.

## 4.2 Per-Object Co-teaching of YOLO Models Using Pseudo Labels

We choose YOLOv5 Jocher et al. (2021) for its excellent speed-accuracy trade-off for downstream tasks. YOLOv5 builds upon the one-stage detection paradigm introduced in the original YOLO Redmon et al. (2016) and incorporates architectural improvements from subsequent versions Redmon & Farhadi (2017; 2018); Bochkovskiy et al. (2020); Jocher et al. (2021); Wang et al. (2022).

We train two separate YOLOv5 models with identical architectures but different initialisations to serve as our co-teaching pair. Both models are optimised for automotive hardware, with particular attention to inference speed and memory footprint.

The core of our approach is a co-teaching mechanism, adapted for object detection, that enables our two student models to learn collaboratively while being robust to the label noise inherent in the teacher's pseudo-labels. Co-teaching was originally proposed for image-level classification tasks with noisy labels Han et al. (2018). However, in object detection, a single image can contain a mix of correctly and incorrectly labelled objects. A simple image-level selection would be suboptimal, as it would discard valuable clean labels within an otherwise "noisy" image.

To address this, we introduce a granular, **anchor-level co-teaching filter**. Coteaching strategy is based on the insight that different network initialisations will cause the two models to learn clean, simple patterns before fitting to the noise in the pseudo labels Arpit et al. (2017). We leverage this by having each model select high-confidence anchor boxes for its peer to train on.

The standard YOLO loss consists of bounding box regression loss ($\ell^{\mathrm{box}}$), classification loss ($\ell^{\mathrm{cls}}$), and objectness loss ($\ell^{\mathrm{obj}}$). We first define a specialised selection loss ($\ell^{\mathrm{sel}}$) to identify clean anchors. We explain this choice in 7.2.

$$\ell_j^{\mathrm{sel}} = \lambda_{\mathrm{box}}\ell_j^{\mathrm{box}} + \lambda_{\mathrm{cls}}\ell_j^{\mathrm{cls}} \tag{4}$$

The filtering process for each mini-batch proceeds as follows:

1. Both student models, $f_\theta$ and $g_\phi$, perform a forward pass on the batch and compute the selection loss $\ell_j^{\mathrm{sel}}$ for every positive anchor.

2. A forget-rate, $r_e$, determines the proportion of anchors to be discarded. We calculate the number of clean anchors to keep, $k = \lceil(1 - r_e)N_{\mathrm{pos}}\rceil$, where $N_{\mathrm{pos}}$ is the total number of positive anchors in the batch.

3. To train model $f_\theta$, we identify the set of anchor indices $\mathcal{K}_f$ that correspond to the $k$ smallest selection losses calculated by its peer, model $g_\phi$.

4. Symmetrically, model $g_\phi$ is given the indices $\mathcal{K}_g$ corresponding to the $k$ smallest selection losses from model $f_\theta$.

Each model is then updated using a masked YOLO loss, where the full detection loss—including the object-ness term—is computed only on the clean set of anchors selected by its peer. The final loss for model $f_\theta$ is:

$$\mathcal{L}_f = \sum_{j \in \mathcal{K}_f} \left( \lambda_{\text{box}} \ell_{f,j}^{\text{box}} + \lambda_{\text{cls}} \ell_{f,j}^{\text{cls}} + \lambda_{\text{obj}} \ell_{f,j}^{\text{obj}} \right) \tag{5}$$

This cross-selection and update strategy creates a robust training loop where each network benefits from the high-confidence selections of the other, effectively filtering out noise and preventing error accumulation.

To further stabilise training, we employ a curriculum learning strategy by gradually adjusting the forget-rate $r_e$ over time. Early in training, the models are still learning basic features, so we use a small forget-rate, meaning we trust a larger portion of the pseudo-labels. As training progresses, the models become more discerning, and we can increase the forget-rate to filter more aggressively. We implement this with a linear ramp-up schedule for the forget-rate:

$$r_e = \hat{r} \cdot \min\left(1, \frac{e}{T_k}\right) \tag{6}$$

Here, $e$ is the current epoch, $T_k$ is a ramp-up period, and $\hat{r}$ is the estimated noise rate of the pseudo-labels. This curriculum allows the models to first learn from a wide distribution of samples and then gradually focus on the cleanest examples, enhancing final model robustness. After training two models in this co-teaching framework, we can use either model for inference or use any ensemble of outputs from both models if required. Complete details of the training methodology are given in Algorithm 1.

## 5 Experimental Setup

**Psuedo Labels**  We discard the ground truth labels from the train dataset and use OWLv2 Minderer et al. (2024) for generating pseudo labels used for training. OWLv2 is the current SOTA foundation model for zero-shot object detection and is widely used for the pseudo-labelling task. For autolabelling, we prompt the model with the name of the class used in the training data. In the results section, GT represents the ground truth labels of the dataset, **Auto Labels** represent the soft outputs from OWLv2 along with the logits without any post-processing. After generating auto labels, we then use Non-Maximum Suppression and Confidence Thresholding with a threshold of 0.3 to generate the pseudo labels. Thus, **Pseudo Labels** represent the hard class labels received from processing the auto labels.

**YOLOv5 as Student Model for Per-Object Coteaching**  For training an efficient downstream model using pseudo labels, we use the YOLOv5 Jocher (2020) architecture. It is a well-studied and widely used model for single-stage object detection. We use YOLOv5 because a full, active and from scratch implementation is publicly available Jocher (2020) with easy access to tuning the model internals. Newer versions of YOLO are not usually published and released as a training API with low-level control Jocher & Qiu (2024). Although we use YOLOv5 for the reasons above, our proposed training procedure is YOLO version-agnostic and can be transferred to other YOLO variants directly. We did not use recent Transformer-based Object Detection method like DeTR Carion et al. (2020) due to higher latency and compute requirements, which defeats our purpose of an efficient real-time detection model. We use the YOLOv5m variant throughout our study. We also note that the coteaching framework will require two models to fit inside a GPU simultaneously. This will mean we will utilise double the GPU memory to fit the same size model and the same batch size during training. We use the per-object coteaching approach proposed in Algorithm 1.

**Baselines**  We compare the performance of our method with the following baselines in a similar setting. For this comparison, we use different approaches with different types of labels and architectures as described below.

- **OWLv2** Minderer et al. (2024): Auto labels generated by the VLM by prompting the model with the name of each class. These Auto labels are evaluated against the ground truth. They are either used as they are or processed to create pseudo labels and used for training in the rest of the approaches.

- **YOLOv5m** Jocher (2020): YOLOv5m model is trained on Pseudo Labels.

| | | | KITTI | | ACDC | | BDD100K | |
|---|---|---|---|---|---|---|---|---|
| *Model* | *Method* | *Labels* | mAP@0.5 | 0.5:0.95 | mAP@0.5 | 0.5:0.95 | mAP@0.5 | 0.5:0.95 |
| OWLv2 | Base | None | 32.34 | 16.52 | 18.82 | 8.92 | 30.81 | 15.86 |
| YOLOv5m | Base | Pseudo | 31.12 | 16.18 | 20.12 | 9.27 | 32.14 | 16.2 |
| YOLOv5m | Soft Distillation | Auto | 34.12 | 17.21 | 21.75 | 10.1 | 35.12 | 16.9 |
| YOLOv5m | Data Distillation | Psuedo | 37.15 | 18.51 | 25.41 | 12.88 | 37.42 | 17.61 |
| Mask R-CNN | Base | Pseudo | 42.21 | 21.12 | 22.43 | 11.72 | 40.13 | 18.88 |
| YOLOv5m | Standard Coteaching | Pseudo | 39.35 | 20.01 | 23.13 | 11.96 | 36.86 | 17.48 |
| YOLOv5m | Per Object Coteaching | Pseudo | **46.61** | **22.05** | **27.31** | **14.28** | **43.21** | **20.81** |
| YOLOv5m | Base | GT | 90.3 | 68.5 | 29.57 | 15.09 | 51.91 | 28.24 |

Table 1: Comparison of different approaches to zero shot object-detection.

- **Soft Distillation** Hinton et al. (2015): YOLOv5m trained with class logits generated from OWLv2.

- **Data Distillation** Radosavovic et al. (2018): Using multiple independent transformations of each image to generate multiple pseudo labels as described in Radosavovic et al. (2018). This set of labels is used for training YOLOv5m.

- **Mask R-CNN** He et al. (2017): Mask R-CNN trained on Pseudo Labels.

- **Coteaching**: Han et al. (2018) Model is trained using vanilla Coteaching by sorting per image loss and discarding a few samples from each mini-batch.

The benchmark performance for our method is a YOLO model trained on the ground truth dataset. In our setting, we are assuming we don't have access to ground truth labels and using VLMs to generate pseudo labels.

**Datasets Used**   We perform experiments on Autonomous driving datasets KITTI Geiger et al. (2012), ACDC Sakaridis et al. (2021), and BDD100k Yu et al. (2020). In all datasets, we used the task of 2D Object detection. KITTI has a training dataset of 7.5k images, which we split into a train and a validation set with an 80:20 ratio. The ACDC dataset contains images from adverse conditions like fog, rain, etc., which are difficult for an autolabeller to label. BDD100k has a total of 70k training set, 10k validation and 20k test set images. We removed labels like 'misc' from the training set of all datasets, as an autolabeller cannot detect such vague labels without specific training data for the label.

**Hyperparameters**   We estimate the optimal noise rate by tuning it for $\hat{r} = 0.0, 0.1, 0.2, 0.4$ and present the results with $\hat{r} = 0.2$. We also present results with all options for comparison in Section 7.1. For training the base YOLO models on pseudo labels, we used the default hyperparameters used by Ultralytics Jocher (2020) ($\lambda_{box} = 0.05$, $\lambda_{cls} = 0.3$ and $\lambda_{obj} = 0.7$) and trained a YOLOv5 from scratch for 200 epochs. For our coteaching approach, we trained the model from scratch for 300 epochs with a noise rate warm-up for 150 epochs, where it increases linearly from 0 to 0.2. We explain the use of linear noise rate warm-up in Section 7.5.

**Performance Metrics Used**   For comparing the performance on Object Detection, we compare the mAP@0.5 and mAP@0.5:0.95 metrics of all the methods for images in the validation set. We also compare the inference efficiency of OWLv2 and YOLOv5m to analyse how viable it is for real-world self-driving deployment.

## 6   Results

In this section, we present detailed experimental evaluations demonstrating the effectiveness of our proposed pipeline.

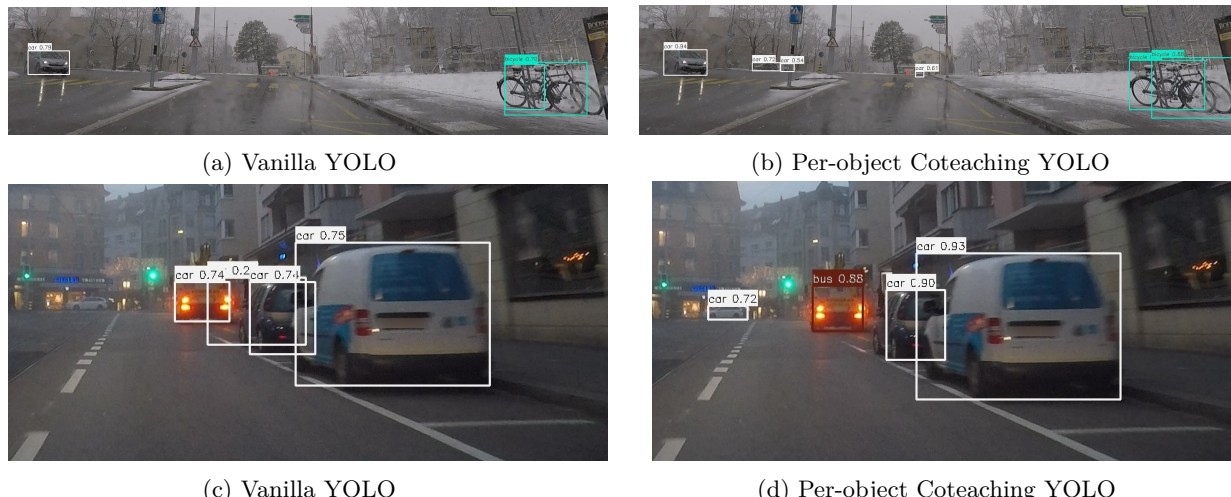

(a) Vanilla YOLO          (b) Per-object Coteaching YOLO

(c) Vanilla YOLO          (d) Per-object Coteaching YOLO

Figure 2: Comparison of predictions made with vanilla YOLO trained and YOLO trained with our method.

## 6.1 Detection Performance

We present the performance comparison of all methods on the validation sets of the KITTI, ACDC and BDD100k datasets in Table 1. We observe clear improvements in detection performance when using our proposed pipeline compared to the baseline distillation method. On the KITTI dataset, our method achieves an mAP@0.5 of 46.61%, substantially higher than the baseline's 31.2%. Similarly, the stricter mAP@0.5:0.95 metric increases from 16.18% in the baseline to 22.05% using our pipeline. We also notice that per-object coteaching outperforms general coteaching performed per-image. The trend is consistent across all three datasets, with comparable relative improvements observed.

We also notice that when the dataset is easier for the model to learn in the presence of ground truth, our pipeline underperforms due to a lack of clean labels. However, in a similar setting in the presence of adverse images (ACDC contains data in adverse conditions like rain, fog, etc.), our pipeline reaches its full potential and performs close to the model trained on ground truth. For example, in a relatively easier dataset like KITTI, the difference in mAP@0.5 with our pipeline and a model trained on ground truth is a massive 43.7%; however, in ACDC, this difference is only 2.26%. We also note that this performance gap of the base VLM generating labels is primarily due to the VLM not detecting the label 'person sitting' as it instead detects them as 'pedestrian'. We did not remove the 'person sitting' class from the pipeline, as ideally, the VLM should be able to distinguish them and detect them accurately.

## 6.2 Efficiency Comparison for Real-Time Object Detection

For real-time object detection applications in autonomous vehicles, computational efficiency and real-time latency are important. We conducted a comprehensive benchmarking of YOLOv5m and OWLv2 (OwlViT-base-patch32) to evaluate their suitability for deployment. We perform our profiling experiments on a single NVIDIA GeForce RTX 2080.

**Inference Performance** YOLOv5m demonstrates significantly superior inference speed with a mean inference time of 12.65ms compared to OWLv2's 38.98ms. This translates to theoretical frame rates of 79.0 FPS for YOLOv5m versus 25.7 FPS for OWLv2. Autonomous driving applications require 30+ FPS for real-time perception. YOLOv5m meets this threshold with substantial headroom, while OWLv2 falls short of real-time requirements.

**Resource Utilization** YOLOv5m contains 21.2M parameters (80.8 MB) compared to OWLv2's 153.2M parameters (584.5 MB). At its peak, YOLOv5m utilises 153.6 MB while OWLv2 utilises 657.3 MB of GPU

memory. For edge deployment scenarios with limited GPU memory, YOLOv5m's 4.3× reduction in GPU memory usage represents a critical advantage.

### 6.3 Predictions on Sample Images

In Figure 2, subfigures a and b contrast the baseline YOLO detector with our per-object co-teaching model on the snowy-weather scene: whereas the baseline a fails to detect three distant cars under low visibility and produces loose, misaligned bounding boxes on the bicycles, our method b recovers all of the cars and fits the bicycle boxes tightly and accurately. Likewise, comparing c and d, the baseline c yields a spurious car detection and misclassifies a bus, but our approach d eliminates that false positive and correctly labels the bus with a tight bounding box. We include more comparisons for reference in Appendix A.

## 7 Ablation study

### 7.1 How to choose Noise rate $\hat{r}$

One considerable limitation of using a coteaching-based strategy for training robust models is requiring a known noise rate $\hat{r}$ for training. We treat the estimated noise rate $\hat{r}$ as a fixed hyperparameter during training and present results with $\hat{r} = 0.2$ in Table 1. In Table 2 we present results with different $\hat{r}$ on all datasets for comparison. We observe that for the ACDC dataset, the detection performance is better when $\hat{r} = 0.4$. This indicates a higher noise rate in the generated pseudo labels for the ACDC dataset.

| $\hat{r}$ | KITTI | ACDC | BDD100k |
|-----|-------|-------|---------|
| 0.0 | 31.1 | 20.14 | 32.11 |
| 0.1 | 41.42 | 24.82 | 38.52 |
| 0.2 | **46.61** | 27.31 | **43.21** |
| 0.4 | 49.24 | **28.54** | 41.27 |

Table 2: Impact of different $\hat{r}$ on detection performance ($mAP@0.5\%$).

### 7.2 Why $\ell^{\mathrm{sel}} = \ell^{\mathrm{box}} + \ell^{\mathrm{cls}}$

We found the objectness score ($\ell^{\mathrm{obj}}$) to be an unreliable signal for selection due to its sensitivity to background in images. Therefore, our selection loss for each positive anchor $j$ deliberately excludes the objectness term. This allows us to identify anchors that are both well-localised and correctly classified, which is a more stable indicator of a clean label. In table 3, we present results of a controlled study with different choices of $\ell^{\mathrm{sel}}$ and that excluding $\ell^{\mathrm{obj}}$ results in better overall performance of the model.

| $\ell^{\mathrm{sel}}$ | mAP@0.5 (%) |
|-----|-------|
| $\ell^{\mathrm{box}}$ | 31.45 |
| $\ell^{\mathrm{cls}}$ | 4.5 |
| $\ell^{\mathrm{obj}}$ | 36.42 |
| $\ell^{\mathrm{box}} + \ell^{\mathrm{cls}}$ | **46.61** |
| $\ell^{\mathrm{cls}} + \ell^{\mathrm{obj}}$ | 34.16 |
| $\ell^{\mathrm{box}} + \ell^{\mathrm{obj}}$ | 31.96 |
| $\ell^{\mathrm{box}} + \ell^{\mathrm{cls}} + \ell^{\mathrm{obj}}$ | 43.82 |

Table 3: Comparision with choice of $\ell^{\mathrm{sel}}$

### 7.3 Increasing unlabeled data

We further analyse the scalability of our proposed pipeline by incrementally varying the amount of unlabeled data used for training the detector. Table 4 illustrates the results of this controlled experiment conducted on the KITTI dataset. We observe a consistent increase in mAP@0.5 performance as we progressively scale up the training set size from 60% to 100% of available unlabeled data.

Specifically, the mAP@0.5 increases from approximately 38% when trained on just 60% of the data, to over 46% with the entire unlabeled dataset. This validates our hypothesis that the co-teaching mechanism effectively filters label noise and allows the model to scale gracefully with additional unlabeled training data. This can be a promising approach, as collecting a lot of unlabeled data is significantly easier compared to labelling existing data precisely.

| Pseudo Labels (%) | Ground Truth (%) | mAP@0.5 (%) |
|---|---|---|
| 60 | 0 | 38.34 |
| 70 | 0 | 39.49 |
| 80 | 0 | 41.98 |
| 90 | 0 | 44.2 |
| 100 | 0 | 46.61 |

Table 4: Impact of increasing the unlabeled images in the pipeline in the KITTI Dataset.

### 7.4 Semi Supervised Setting

We mix some percentage of ground truth labels in the training data and analyse how this affects our performance. As we increase the GT data, especially in datasets with huge differences in performance when trained on ground truth, the performance increases significantly. This is mainly due to some labels that the VLM couldn't pick up during the labelling process, but a few samples from the ground truth significantly improved the model's ability to identify and detect these classes. We present the results in table 5. We show that incorporating just 10% of precisely labelled ground truth data improves the performance from 46.61% to 57.97%.

| Pseudo Labels (%) | Ground Truth (%) | mAP@0.5 (%) |
|---|---|---|
| 100 | 0 | 46.61 |
| 95 | 5 | 49.42 |
| 90 | 10 | 57.97 |
| 85 | 15 | 65.13 |
| 80 | 20 | 72.42 |
| 75 | 25 | 77.80 |

Table 5: Impact of incorporating a small percentage of ground truth annotations during training on the KITTI Dataset.

### 7.5 Impact of curriculum learning

Using a curriculum learning based strategy by linearly increasing the noise rate is known to improve coteaching-based methods. This is helpful for learning as the model can learn easy patterns in early epochs without dropping too many samples, even if they're noisy. As the model trains, it starts memorising noise from the training data, so it helps to drop more noisy labels from training. This also helps with stable training of the model. If we directly use the noise rate from epoch one, we leave out some useful data, leading to model underfitting. In our application, training YOLOv5m on KITTI data with pseudo labels and $\hat{r} = 0.2$ for 300 epochs without linear warm up results in $mAP@0.5 = 41.78$, whereas linearly increasing $\hat{r} = \{0, 0.2\}$ from epochs 1 to 150 and training for 300 epochs results in $mAP@0.5 = 46.61$.

# 8 Conclusion

**Limitations and Future Work**   One of the limitations of our approach is requiring a known noise rate for training. This is not practical as estimating the noise rate in pseudo labels is non-trivial. Estimating the noise rate online as part of the training algorithm can be a potential extension. Our approach is limited to 2D bounding boxes, while Autonomous driving systems now largely rely on 3D bounding boxes. Extending the per-object coteaching style method for 3D Object Detection models like VoxelNet Qi et al. (2019) will get it closer to deploying in current systems. We used the name of the class to prompt the VLM for generating pseudo labels. Using advanced prompting methods and ensemble-based strategies with the VLM can improve the quality of the pseudo labels used for training.

**Conclusion**   Our comprehensive evaluations demonstrate the clear advantages of our proposed pipeline. Our per-object co-teaching mechanism robustly addresses pseudo-label noise, significantly improving accuracy across multiple datasets and evaluation metrics compared to baseline distillation. Additionally, our pipeline maintains efficient real-time inference, which is vital for practical autonomous driving applications. We also illustrate that the judicious use of even minimal ground truth labels or increased unlabeled data can both substantially boost performance, highlighting our method's practical viability in real-world autonomous driving scenarios.

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

## A Comparison

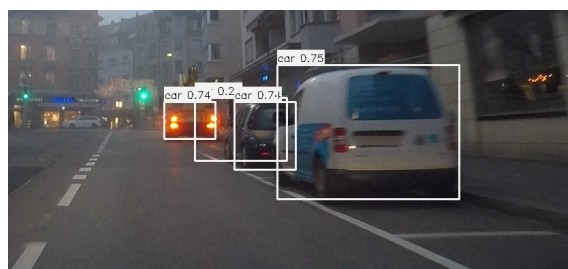 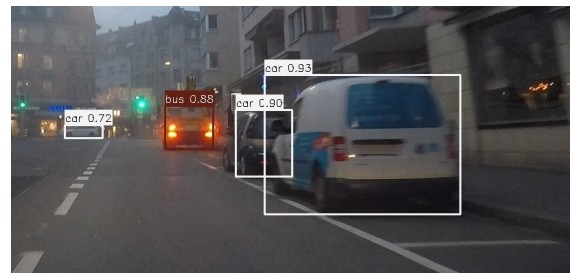

(a) Base YOLO                                    (b) Our Method

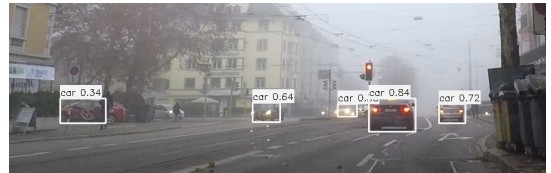 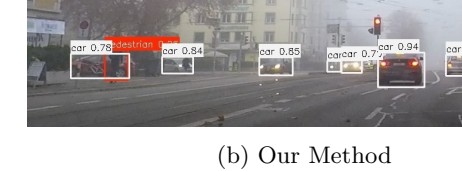

(a) Base YOLO                                    (b) Our Method

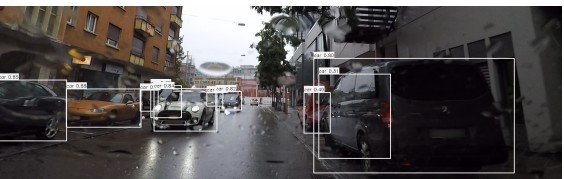 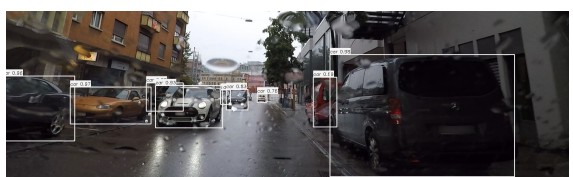

(a) Base YOLO                                    (b) Our Method

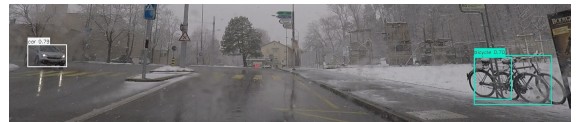 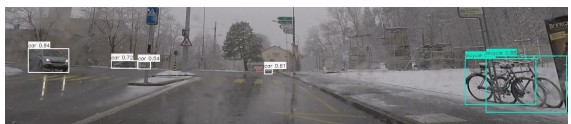

(a) Base YOLO                                    (b) Our Method

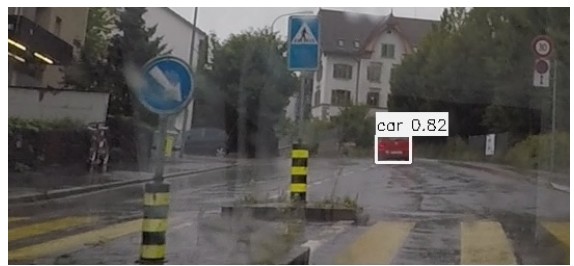 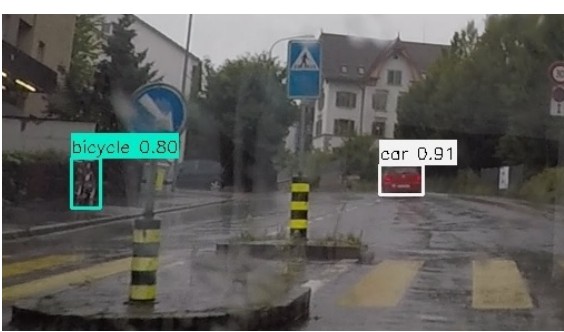

(a) Base YOLO                                    (b) Our Method

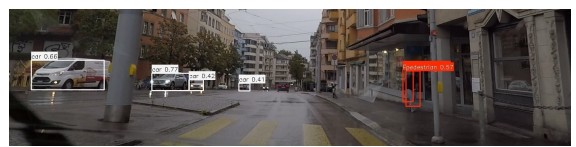 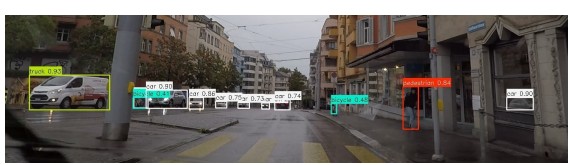

(a) Base YOLO                                    (b) Our Method

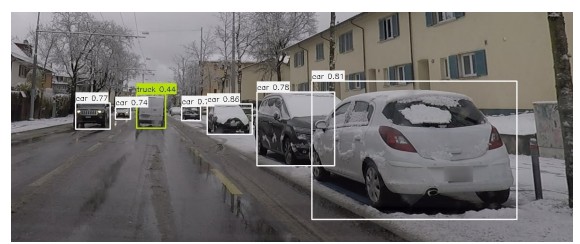 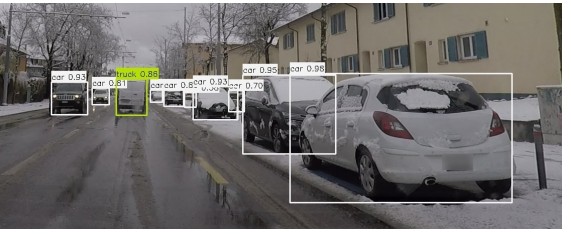

(a) Base YOLO                                    (b) Our Method

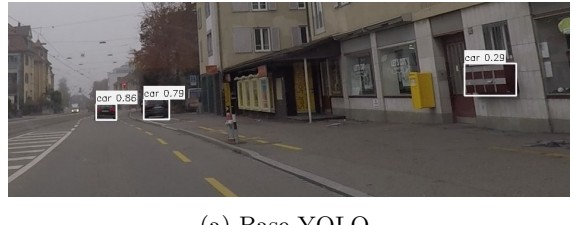
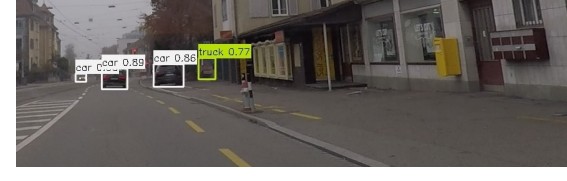

(a) Base YOLO                                        (b) Our Method

Figure 10: Comparison of predictions made with vanilla YOLO trained and YOLO trained with our method.

