# OpenReview forum: "Robust Object Detection with Pseudo Labels from VLMs using Per-Object Co-teaching"
_TMLR — Rejected by TMLR_

### Review · Reviewer_eUUy · 2025-12-15

**Summary Of Contributions:**

- This paper proposes a framework to train lightweight, real-time object detection models (YOLOv5m) using pseudo-labels generated by a large-scale Vision-Language Model (VLM). To mitigate the noise inherent in VLM-generated labels, the authors introduce an object-level (anchor-level) co-teaching strategy. Experimental results on the KITTI, ACDC, and BDD100k datasets demonstrate that the proposed method outperforms baselines trained with simple pseudo-labeling or standard image-level co-teaching.
- Strengths
  - The paper integrates existing robust learning techniques (co-teaching) with the emerging paradigm of VLM-based distillation, demonstrating a practical approach to leveraging unlabeled data.
  - The proposed method achieves better performance compared to standard baselines on multiple datasets, showing the potential of filtering noisy pseudo-labels at the object level.
- Weaknesses
  - The core technical contribution claimed by the authors, per-object co-teaching, is not novel. Chadwick and Newman (2019) have already addressed noisy labels in object detection using per-object co-teaching. The primary difference is the switch from R-CNN to YOLO, which is an incremental application. The paper incorrectly frames this as a novel contribution.
  - The paper claims that combining VLM pseudo-labels with robust learning yields good object detectors, but the depth of analysis is lacking. For instance, there is a massive performance gap between the proposed method (mAP 46.61\%) and the model trained on ground truth (mAP 90.3\%) on the KITTI dataset. The paper does not investigate the root causes of this gap (e.g., is it due to the recall limit of the VLM, or over-filtering by the co-teaching mechanism, etc.).
  - Critical details required for reproducibility and understanding are missing. The hyperparameter $\hat{r}$ is central to the method, yet there is no analysis or explanation of how it is determined. Furthermore, the descriptions of baselines (e.g., Soft Distillation, Data Distillation) are too vague to understand the exact implementation.

**Audience:**

No

**Audience Explanation:**

- The proposed “per-object co-teaching” is largely a re-implementation of Chadwick and Newman (2019) applied to a YOLO architecture. The paper does not offer new scientific insights into the nature of VLM noise or why the proposed method falls significantly short of GT performance. Without deep analysis, the finding that “robust learning helps with noisy labels” is an expected result rather than an interesting finding for the community.

**Claims And Evidence:**

No

**Claims Explanation:**

- While the authors present empirical results showing performance gains, the evidence is not sufficiently “clear” due to significant ambiguity in the experimental setup.
- The manuscript does not explain how the estimated noise rate $\hat{r}$ is calculated or selected. Since co-teaching is highly sensitive to this parameter, omitting this detail makes the evidence unconvincing.
- The implementation details for baselines like “Soft Distillation” and “Data Distillation” are not provided. Without knowing the exact settings, it is difficult to verify if the comparisons are fair and accurate.
- Terms like “Hard labels” are introduced in Section 5 but are not consistently defined or used, adding to the confusion.

**Requested Changes:**

- The authors must explicitly acknowledge Chadwick and Newman (2019) as the origin of the per-object co-teaching strategy. The contribution should be redefined not as proposing a new method, but as applying/adapting this strategy to the specific context of VLM-distillation for single-stage detectors (if a distinct technical modification exists, it must be clearly highlighted).
- The paper needs a more rigorous analysis of the results. Specifically, the authors should investigate the large performance gap between the proposed method and the GT baselines (Table 1). An analysis of the noise characteristics (e.g., class-specific quality of VLM pseudo-labels) and the impact of the curriculum learning strategy is necessary to provide value to the reader.
- Provide a clear explanation of how the noise rate $\hat{r}$ is determined.
- Clearly describe the implementation details of all baselines (Soft/Data Distillation).
- Fix inconsistent terminology (e.g., define “Hard labels” clearly and use it consistently)
- Ensure citations are contextually appropriate. For example, citing co-teaching paper when discussing the general unreliability of VLMs (Introduction, paragraph 2) or citing Chadwick and Newman (2019) for computational cost arguments is inappropriate. They should be cited in the context of the methodology used.

---

> ### Author Response · Authors · 2026-01-05
>
> To reviewer eUUy,
>
> We appreciate the detailed review with crucial suggestions to improve our work. We addressed all the concerns raised and requested changes in our work. We request that the reviewer read our updated manuscript, as we have incorporated all the requested work in this edition. We briefly explain our changes below and address some raised concerns.
>
> - **Novelty:** We agree that our core idea is inspired by [1] and adapted to YOLO Architecture using anchor level selection and a selection loss that excludes objectness score. We also focus on the specific application of training zero-shot detectors using pseudo-labels for Autonomous Driving. We changed the framing in our paper's abstract and introduction section to better reflect this and present our work as an extension of [1].
>
> - **Experimental Setup:** We rewrote the experimental setup with sufficient details and clarity to better describe our setting. We added a detailed description of how labels are extracted and baselines are trained. We hope this can remove any ambiguity around our experiments.
>
> - **Ablation Study:** In our setting, we assume the noise rate is known, which is standard in coteaching literature [2] and use $\hat r = 0.2$ for presenting our results. However, we added ablations in Section 7.1 with results on how $\hat r = \{0.0, 0.1, 0.2, 0.4\}$ impacts the model performance. We also address this as a limitation in the conclusion of our paper. We also added a description of why the curriculum learning strategy is required and how it affects practical performance in our pipeline. We also add more output comparison with our approach in the Appendix.
>
> - **Performance gap for KITTI:** We note that this performance gap of the VLM is primarily due to the VLM not detecting the label ‘person sitting’ as it instead detects them as ‘pedestrian’. We have removed classes like ‘misc’ from the label space due to ambiguity for the model. However, we did not remove the ‘person sitting’ class as the VLM should be able to distinguish them and detect them accurately. Our core contribution is that, despite the quality of the pseudo labels used for training, our pipeline can improve the performance of YOLO compared to the model that generates the labels. We also added this reasoning in the manuscript for better clarity.
>
> - **Appropriate citations:** We apologise for the simple errors in citations and tried to cite appropriate citations for the context throughout the paper.
>
> [1] Chadwick, Simon, and Paul Newman. "Training object detectors with noisy data." In 2019 IEEE Intelligent Vehicles Symposium (IV), pp. 1319-1325. IEEE, 2019.
>
> [2] Han, Bo, Quanming Yao, Xingrui Yu, Gang Niu, Miao Xu, Weihua Hu, Ivor Tsang, and Masashi Sugiyama. "Co-teaching: Robust training of deep neural networks with extremely noisy labels." Advances in neural information processing systems 31 (2018).

---

### Review · Reviewer_EanW · 2025-12-16

**Summary Of Contributions:**

This paper addresses the challenge of training real-time object detectors for autonomous driving with limited labeled data by leveraging Vision-Language Models (VLMs), specifically OWLv2, for zero-shot pseudo-labeling. To mitigate the issues of high inference latency and label noise (hallucinations) inherent in VLMs, the authors propose a novel "per-object co-teaching" framework. In this setup, two YOLOv5 student models are trained collaboratively; instead of filtering data at the image level, they selectively filter out unreliable bounding boxes (anchors) for each other based on loss values.

**Audience:**

No

**Audience Explanation:**

Although the work is framed for autonomous driving, the focus solely on 2D bounding boxes limits its practical value. Since modern autonomous driving systems rely heavily on 3D object detection for depth estimation and planning.

**Claims And Evidence:**

Yes

**Claims Explanation:**

1. The advantage of employing a VLM like OWLv2 as a teacher is not evident, particularly given the low baseline detection quality reported (e.g., 18.82% mAP on ACDC). The authors should explicitly evaluate the raw detection quality of the VLM to justify its use; if the quality is poor, it is unclear why a VLM is a better teacher than a specialized detector trained on existing data.

2. The claims regarding "efficient, real-time object detection" are misleading, as the reported speed and low latency are inherent attributes of the YOLOv5m architecture  rather than the proposed per-object co-teaching method. The manuscript should clarify that the performance gains stem from the model selection itself, not the training pipeline.

3. Although the work is framed for autonomous driving, the focus solely on 2D bounding boxes limits its practical value. Since modern autonomous driving systems rely heavily on 3D object detection for depth estimation and planning, the authors should address the limitations of a 2D-only approach in this context.

4. The paper overlooks key comparisons with representative semi-supervised learning methods, which is critical given the reliance on unlabeled data. The authors should discuss and compare their approach with relevant state-of-the-art works, specifically UniMatch V2 (IEEE TPAMI), Leveraging Unlabeled Data to Boost Oriented Object Detection (IEEE TPAMI), and Sparse Semi-DETR (CVPR 2024).

**Requested Changes:**

1. The advantage of employing a VLM like OWLv2 as a teacher is not evident, particularly given the low baseline detection quality reported (e.g., 18.82% mAP on ACDC). The authors should explicitly evaluate the raw detection quality of the VLM to justify its use; if the quality is poor, it is unclear why a VLM is a better teacher than a specialized detector trained on existing data.

2. The claims regarding "efficient, real-time object detection" are misleading, as the reported speed and low latency are inherent attributes of the YOLOv5m architecture  rather than the proposed per-object co-teaching method. The manuscript should clarify that the performance gains stem from the model selection itself, not the training pipeline.

3. Although the work is framed for autonomous driving, the focus solely on 2D bounding boxes limits its practical value. Since modern autonomous driving systems rely heavily on 3D object detection for depth estimation and planning, the authors should address the limitations of a 2D-only approach in this context.

4. The paper overlooks key comparisons with representative semi-supervised learning methods, which is critical given the reliance on unlabeled data. The authors should discuss and compare their approach with relevant state-of-the-art works, specifically UniMatch V2 (IEEE TPAMI), Leveraging Unlabeled Data to Boost Oriented Object Detection (IEEE TPAMI), and Sparse Semi-DETR (CVPR 2024).

---

> ### Author Response · Authors · 2026-01-05
>
> To reviewer EanW,
>
> We appreciate the reviewer's detailed analysis of our work and have addressed the requested changes. We request the reviewer to read our updated manuscript and evaluate accordingly. We also addressed the concerns raised by the reviewer below.
>
> - **2D Bounding Box Limitation:** We acknowledge this limitation of our approach and added it to our manuscript.
>
> - **OWLv2 as teacher model:** The core contribution of our work is to train object detection models using pseudo labels generated from VLMs. We assume no existing data for training a specialised model. We propose a pipeline to train the model by generating pseudo labels from the VLM. We also reported the raw detection quality of the VLM in Table 2. The reason our approach outperforms the labels that it uses to train on is that our approach discards noisy samples online from the training loop, learning better representations from clean samples.
>
> - **Misleading claims:** The reviewer is right to point out that the efficient real-time object detection claims primarily stem from the model choice and not our proposed pipeline. Our contribution is primarily a coteaching-based training method to train robust models, and we use YOLOv5m as the student model due to its efficient latency. We update the manuscript to reflect this claim.
>
> - **Better Baselines:** Our work is also focused on zero-shot detection from pseudo labels, and a semi-supervised setting was just an ablation added for better analysis of the algorithm. We included recent and relevant baselines of our setting to the best of our ability. We also added a baseline training Mask R-CNN [1] on pseudo labels upon the reviewers' request.
>
> [1] He, Kaiming, Georgia Gkioxari, Piotr Dollár, and Ross Girshick. "Mask r-cnn." In Proceedings of the IEEE international conference on computer vision, pp. 2961-2969. 2017.

---

### Review · Reviewer_jzgB · 2025-12-21

**Summary Of Contributions:**

This paper proposes a pipeline for training real-time object detectors (yolov5m) using pseudo-labels generated by large vision-language models (VLMs). Specifically, the authors use OWLv2 to produce noisy pseudo bounding boxes and train a lightweight YOLOv5 detector via a per-object co-teaching strategy. Two YOLO models are trained jointly, where each model filters unreliable pseudo-labeled objects (at the anchor level) for its peer based on per-object loss values, instead of discarding entire images. The goal is to mitigate noise in VLM-generated pseudo-labels while achieving real-time inference performance suitable for autonomous driving. The method is evaluated on KITTI, ACDC, and BDD100K datasets, showing improved mAP over several baselines trained on raw or filtered pseudo-labels.

**Strengths**:

1. The paper addresses a relevant and practical problem: how to leverage powerful but slow and noisy VLM-based detectors to train efficient real-time object detectors suitable for deployment.

2. The proposed method is simple to follow.

**Weaknesses**:

1. The related work section does not adequately discuss or contrast the proposed method with closely related prior approaches, particularly earlier VLM-based pseudo-labeling and semi-supervised detection frameworks. This weakens the paper’s positioning.

2. While improvements over selected baselines are shown, the evaluation lacks stronger or more recent comparison methods (baselines are proposed as 2018 or earlier).

**Audience:**

No

**Audience Explanation:**

While the topic is relevant, the paper is unlikely to generate strong interest because: 1. the core idea closely follows established pseudo-labeling and semi-supervised detection paradigms; (2) the experimental evaluation is not sufficiently extensive to convincingly position this work as a meaningful step beyond prior art.

**Claims And Evidence:**

No

**Claims Explanation:**

Main claims made by the paper:

1. Per-object co-teaching effectively mitigates noise in VLM-generated pseudo-labels, leading to better detection performance.

2. The proposed pipeline enables scalable training of real-time object detectors without heavy reliance on human annotations, and can be further improved with a small amount of ground-truth data.

These claims are partially supported.

The experimental results do show consistent performance improvements over a baseline YOLOv5 model trained on pseudo-labels and over several common distillation methods. However, the paper does not convincingly demonstrate that the proposed approach represents a technical advance over existing pseudo-labeling or semi-supervised object detection methods leveraging VLM. In particular, similar pipelines: leveraging large pretrained models to generate noisy pseudo-labels and training efficient detectors using filtering, self-training, or semi-supervised learning, have been extensively studied. Prior works [1] already demonstrate conceptually similar strategies using VLM-generated pseudo-labels combined with semi-supervised learning. The paper does not provide a clear distinction between its method and these prior approaches.

[1] Zhao, Shiyu, et al. "Exploiting unlabeled data with vision and language models for object detection." ECCV, 2022.

**Requested Changes:**

1. Expand the related work section: (1) explicitly discuss prior VLM-based pseudo-labeling and semi-supervised object detection methods, (2) Clearly articulate how the proposed method differs technically and conceptually from these approaches.

2. Strengthen experimental evaluation: including more challenging comparisons (VLM-based pseudo-labeling / semi-supervised leanring...).

3. Improve positioning: better justify why this approach should be preferred over existing VLM-based semi-supervised alternatives.

---

> ### Author Response · Authors · 2026-01-05
>
> To reviewer jzgB,
>
> We appreciate the detailed review and analysis of our work. We addressed all the mentioned concerns, requested changes and added them to our manuscript. We request the reviewer to read our updated manuscript and evaluate the changes. We also briefly address the concerns raised below.
>
> - **Incomplete Related Work:** We notice we missed out some important relevant works from our submission. We updated the manuscript with the mentioned paper and more important works in relevant domains, with a brief explanation and comparison when required.
>
> - **Strong Baselines:** We tried to include recent and relevant baselines to the best of our ability. As this is a fairly new paradigm, there aren’t many works dealing with similar settings. In the previous work mentioned [1] and other similar works like [2], the pseudo-label generation pipeline is significantly different, resulting in different training data and cannot be a fair comparison. Our work is also focused on zero-shot detection from pseudo labels, and the semi-supervised setting was just an ablation added for better analysis. We added a baseline training Mask R-CNN [3] on pseudo labels at the reviewers' request.
>
> - **Experimental Analysis:** We added one more baseline and stronger ablation analysis to better convey the significance of our method and its improvements beyond existing approaches. We also add more output comparison with our approach in the Appendix.
>
> [1] Zhao, Shiyu, Zhixing Zhang, Samuel Schulter, Long Zhao, B. G. Vijay Kumar, Anastasis Stathopoulos, Manmohan Chandraker, and Dimitris N. Metaxas. "Exploiting unlabeled data with vision and language models for object detection." In European conference on computer vision, pp. 159-175. Cham: Springer Nature Switzerland, 2022.
>
> [2] Gao, Mingfei, Chen Xing, Juan Carlos Niebles, Junnan Li, Ran Xu, Wenhao Liu, and Caiming Xiong. "Open vocabulary object detection with pseudo bounding-box labels." In European Conference on Computer Vision, pp. 266-282. Cham: Springer Nature Switzerland, 2022.
>
> [3] He, Kaiming, Georgia Gkioxari, Piotr Dollár, and Ross Girshick. "Mask r-cnn." In Proceedings of the IEEE international conference on computer vision, pp. 2961-2969. 2017.

---

### Decision · Action_Editor_nGZD · 2026-02-12

**Recommendation:** Reject

**Audience:**

No

**Audience Explanation:**

The reviewers note that, although the topic is relevant, the paper is unlikely to interest a meaningful subset of the TMLR audience because it mainly shows a reasonable combination of known components beating weak/outdated baselines, without extracting generalizable insights. They argue the evaluation does not sufficiently contrast with the most relevant recent methods and the results read more like a case study of one pipeline than findings that broadly teach the community something useful.

**Claims And Evidence:**

No

**Claims Explanation:**

The reviewers agree that the paper’s claims are not supported by sufficiently convincing evidence. The current evaluation is limited to a 2D-only detector pipeline, yet the paper frames the work in an “autonomous driving” context, which overstates practical relevance given what is actually demonstrated. The reviewers perceive the experimental validation as incomplete: it omits comparisons to strong, representative recent baselines (including relevant methods previously suggested by reviewers), making it difficult to assess whether the observed gains reflect a meaningful advance or merely an improvement over weak/outdated references. The claimed broader takeaway about VLM-generated pseudo-labeling pipelines is not convincingly established: the method largely follows known patterns and the reported improvements are consistent with expected benefits of robust learning under label noise.

**Resubmission Of Major Revision:**

The authors may consider submitting a major revision at a later time.